# Effectiveness of Global Postural Reeducation in Ankylosing Spondylitis: A Systematic Review and Meta-Analysis

**DOI:** 10.3390/jcm9092696

**Published:** 2020-08-20

**Authors:** Gloria Gonzalez-Medina, Veronica Perez-Cabezas, Antonio-Jesus Marin-Paz, Alejandro Galán-Mercant, Carmen Ruiz-Molinero, Jose Jesus Jimenez-Rejano

**Affiliations:** 1Department of Nursing and Physiotherapy, University of Cadiz, 11009 Cadiz, Spain; gloriagonzalez.medina@uca.es (G.G.-M.); antoniojesus.marin@uca.es (A.-J.M.-P.); alejandro.galan@uca.es (A.G.-M.); carmen.ruizmolinero@uca.es (C.R.-M.); 2Department of Physiotherapy, University of Seville, 41009 Seville, Spain; jjjimenez@us.es

**Keywords:** ankylosing spondylitis, global postural reeducation, dysfunction, range of motion, pain

## Abstract

The aim of this study was to determine the role of global postural reeducation for people with ankylosing spondylitis. We compared the effects of treatments on pain, dysfunction (using the Bath Ankylosing Spondylitis Disease Activity Index and Bath Ankylosing Spondylitis Functional Index), range of motion, and chest expansion in a specific population aged over 18 years old with ankylosing spondylitis. We performed a systematic review and meta-analysis following the Preferred Reporting Items for Systematic Reviews and Meta-Analysis (PRISMA) statements. The search was conducted using the PubMed, Physiotherapy Database (PEDro), Scientific Electronic Library Online (SciELO), and Web of Science (WoS) databases. Clinical trials and systematic reviews/meta-analysis were reviewed. Results: 154 studies were found. Finally, four were included. Conclusions: global postural reeducation is beneficial for ankylosing spondylitis, but no more so than other conventional treatments, except for spinal mobility, where Global Postural Reeducation demonstrated an advantage.

## 1. Introduction

Ankylosing spondylitis (AS) is a specific spondyloarthritis that affects the axial skeleton and has extra-axial manifestations, such as uveitis or psoriasis. AS occurs predominantly in young males and is characterized by pain and stiffness of joints (ankylosis) with inflammation at tendon insertions [1,2]. At the level of the spine, there is vertebral bodies fusion, which is the so-called “bamboo spine” [3]. In the chest, rigidity of the chest wall and persistent immobility appear, which can affect respiratory function [4,5].

This pathology is characterized by a strong genetic association between the *HLA* B27 allele. AS has been reported but other genes are also involved in the so-called *IL*-17 *A/IL*-23 inflammatory immune axis. There are also molecular polymorphisms that determine the occurrence of the disease [1].

Although there is consensus that the average age of onset of this condition is 40 [2], its incidence and prevalence rates are variable depending on the country. Bohn et al. [6] indicated incidence ranges from 0.4 to 15 per 100,000 population (Iceland vs. Ontario, Canada) and the prevalence from 6.5 to 500 per 100,000 (Japan vs. Turkey). Smith [2] indicated that, in the United States, the prevalence was approximately 0.5%.

There is also agreement on the conservative treatment, which consists of medicines, including tumor necrosis factor (TNF) antagonists (etanercept, infliximab, golimumab, and adalimumab) [7]. These have been shown, in the short and long term, to be effective treatments for AS [7,8]. In addition, there are other types of treatments that complement these medicines. Therapeutic physical exercise is a complementary treatment option, which can help to reduce the symptoms, especially when focused on the spine mobility and peripheral joints [9]. We focused our research on global postural reeducation (GPR). GPR is a method of clinical intervention that was developed by Philippe Souchard in the 1950s. This method is based on the principles of individuality, causality, and globality [10]. In addition, it uses active stretching of the “muscle chains” by performing specific postures. The GPR method includes five muscular chains: the dorsal chain, brachial chain, anterior chain of the neck, antero-lumbar chain, and respiratory chain. The use of these postures aims to correct the existing retractions in the different muscular chains [11]. This type of treatment should be focused on the results obtained in the evaluation of the patient, and needs to be prescribed individually [12].

Previous studies have shown the effectiveness of GPR treatment to be superior to physical therapy [12], and, according to the literature, the effects of GPR are quite broad in other pathologies, acting on muscle flexibility [9,13,14,15,16,17,18,19], postural organisation [19,20,21,22], functionality [6,23,24], cognition, quality of life [6,16,21,25] and reducing pain [16,17,18,26], fatigue [24], and others [21,27,28,29]. The benefits of GPR in AS, however, have been analysed either in conjunction with other types of treatment or were not studied in depth [4,30,31].

Therefore, the main aim of this study was to determine the efficacy of GPR as a conservative treatment for people with AS. The secondary endpoints were: (1) to assess the effect of GPR on the disease activity, functionality, mobility, pulmonary, and aerobic functions in AS; and (2) to establish the specific benefits on the selected outcomes, to evaluate if GPR is an effective treatment option in adults diagnosed with AS.

## 2. Material and Methods

### 2.1. Study Design

We performed a systematic review and meta-analysis using the Preferred Reporting Items for Systematic Reviews and Meta-analysis (PRISMA) [32]. The review was planned and conducted following the PRISMA checklist (Appendix A). In this review, we included randomized clinical trials and systematic reviews. The bibliographic search took place from December 2019 to March 2020 and included the PubMed, Physiotherapy Database (PEDro), Scientific Electronic Library Online (SciELO), and Web of Science (WoS) databases.

### 2.2. Search Strategy

The search of the literature for the present review is detailed in Table 1. Filters for publication dates or language were not applied. We found a total of 149 potential articles. Study type filters were used, selecting only clinical trials and reviews, studies conducted in humans, and studies whose participants were over 18 years of age. Studies published in Korean, clinical trials without control groups, and those in which the main pathology was not spondylitis were eliminated. The process of obtaining the results, screening, and eligibility is described in the flowchart. Two reviewers screened the full articles to determine which to include in the review and quantitative and qualitative analysis.

### 2.3. Criteria for Considering Studies for This Review

Studies included in this review met the following inclusion criteria (PICO): (a) the participants were adults diagnosed with AS; (b) a physical intervention was performed according to the GPR method; (c) they were compared to other types of therapy or non-intervention; (d) the outcomes were related to an improvement in the symptomatology of AS (mainly, our results were related to the disease activity, functionality, mobility, and pain intensity); and (e) the study design was a randomized controlled trial.

### 2.4. Data Extraction

Two researchers independently reviewed and systematically extracted the data from each study and arrived at a consensus on all items. The following information was extracted from the studies: for the clinical trials, the general data that were checked included: the bibliographic data (authors, journal, year of publication, and database(s)); study characteristics (topic, aim of study, study design, sample size, groups (numbers of groups and size), drop out, and randomized (yes/no)); characteristics of participants (gender, age, symptoms, and characteristics); description of the intervention (both intervention and control groups, number of treatment sessions, duration of each session, and total treatment time); outcomes assessed; study results; and digital object identifier (DOI). The information was obtained with an in-depth and impartial reading of the studies. A third researcher was consulted in case of doubts or discrepancies between the first two researchers. The missing data were not consulted by the authors.

### 2.5. Data Analysis and Outcomes

The quantitative synthesis of the results (meta-analysis) was conducted by another author. The data were pooled if at least two studies presented similar and comparable outcomes. A meta-analysis was performed to determine the effect of GPR. The disease activity (Bath Ankylosing Spondylitis Disease Activity Index, BASDAI [33,34]), functionality (Bath Ankylosing Spondylitis Functional Index, BASFI [33,35]), mobility (Bath Ankylosing Spondylitis Metrology Index, BASMI [34]; cervical rotation, chest expansion, finger-floor distance, and modified Shöber test); and pain (Visual Analog Scale, VAS) were measured. In addition, the changes were compared in the effect size (pre-intervention and post-intervention) between the intervention group and the control group. The standardised mean difference was calculated along with the 95% confidence interval, with the significance level set at p < 0.05. The statistical analyses were performed with the statistical software Review Manager 5.3 (The Nordic Cochrane Centre, The Cochrane Collaboration, 2014).

Whenever possible, the publication bias was estimated using the Begg and Egger [30] tests and a funnel plot. If applicable, a sensitivity analysis was performed to estimate the degree of influence of each article included in each meta-analysis on the results of that meta-analysis. To check for heterogeneity, Galbraith’s graph was used. On one hand, in the case of heterogeneity between studies (p < 0.01), a random effects model was used in the analysis and the corresponding forest plot was reported. On the other hand, a fixed-effect model was used in the case of homogeneity. Pulmonary functions could not be analysed by meta-analysis and were described qualitatively.

### 2.6. Evaluation of Clinical Relevance

To assess the quality of the evidence of the outcomes presented in the reported trials, GRADE (Grading of Recommendations Assessment, Development, and Evaluation) was performed [36]. GRADE consists of the following five items: risk of bias, inconsistency, indirectness, imprecision, and other considerations. Each domain was defined as not serious, serious, or very serious. The resulting quality assessment of the evidence was classified as high, moderate, low, or very low.

The robustness of the evidence was assessed using the Grade Development Tool (GDT). This was classified as high, moderate, low, and very low. The degree of recommendation (strong or weak recommendation) was reported separately. This will provide a clear and pragmatic interpretation of the degree of recommendation (strong or weak) for clinicians, patients, and managers. The importance to patients of the outcome variables of the therapeutic alternatives considered was assessed. The methodological quality of the studies was evaluated with the PEDro scale. Two independent blinded researchers selected the studies. Subsequently, a discussion took place for the final selection, and the arithmetic mean was used to obtain the final results.

## 3. Results

The number of studies screened, assessed for eligibility, and included in the review; with reasons for exclusions, is shown in the flow chart (Figure 1).

### 3.1. Data Extraction

The samples of the studies analysed comprised a greater number of men (75.58%) than women (24.42%), and the ages of the participants ranged from 33.30 (± 10.49) to 44.27 (± 10.55). The number of groups to which participants were assigned varied. In the Coksevim’s and Durmus´s trials [24,37], there were three groups, and in the Silva and Fernández de las Peñas trials [4,28], there were two.

The GPR was the intervention program in all studies (Table 2), with differences in its application. Only in one of the studies [28], did participants receive supplementary treatment: a 30-minute exercise, five days a week for three months. All other studies gave a postural care guide [4] with instructions not to change their lifestyle [28]. In the Durmus´s study [37], nothing was indicated regarding supplementary treatment. The GPR intervention in two of the studies [28,37] was exactly the same: a general and specific warm up, dynamic axial exercise, static postural exercise, specific breathing exercises, and cool down. The only difference was in Coksevim’s research [24], in which participants were simultaneously treated with anti-TNF (Anti- tumour necrosis factor).

Follow up was very similar, ranging from three [24,37] to four months [4,28]. Instructions on how to perform conventional exercises, applied to the control group, were given to participants in three of the four clinical trials [4,24,37]. These consisted, in two cases [28,37], of written step-by-step instructions with illustrations; and in one of the studies, it was only stated that no further physiotherapy intervention should be received during the study [28].

### 3.2. Data Analysis and Meta-Analyses

Seven different meta-analyses were performed. In the disease activity, functionality, and mobility; the results of the BASDAI, BASFI, cervical rotation, chest expansion, finger-floor distance, and modified Shöber test were considered as variables. With regard to pain, the amount of pain referred by VAS was studied. In relation to the results variables, we observed that there was a certain consensus in the measurement tools. BASDAI and BASFI were used by all authors, although in the case of the Fernández de las Peñas study they were used in a disaggregated way. The modified Shöber test and VAS were used in three studies. Cervical rotation, chest expansion, and finger-floor distance were used in only two of the three studies.

The results obtained in the quantitative synthesis of the studies in the meta-analysis for the variables are shown in Figure 2. In the meta-analyses, under the random effects model, we found that there was no significant difference between the control group and the experimental group, except for the finger-floor distance. A significant effect on the assessment of total mobility was observed when leaning forward in a standing position (mean difference d = −3.81 Confidence Interval (95%): −4.81; −2.81]). That is, the total mobility when leaning forward was significantly better in the experimental group compared to the control group (Figure 2e).

### 3.3. Risk of Bias, Sensitivity and Heterogeneity

There is no statistical evidence of the existence of publication bias according to the results offered by the Begg and Egger tests (*p* > 0.05 This is also shown in the following funnel plots (Figure 3). The sensitivity analysis indicated that no study substantially modified the overall results when eliminated.

In order to investigate whether heterogeneity exists, the Galbraith graph was used and we observed that, in this meta-analysis, there was great heterogeneity among the studies (*p* < 0.01); thereby, a random effects model was used in these analyses (Figure 4).

A spirometer was used to measure lung function. Forced expiratory manoeuvres were performed from maximum inspiration for FEV1 (forced expiratory volume in 1 second), FVC (forced vital capacity), and PEF (peak expiratory flow). Slow tests were performed for VC (vital capacity) and MVV (maximum voluntary ventilation). We found that GPR produced improvements in lung function in the FVC, FEV1, and PEF parameters.

### 3.4. Evaluation of Clinical Relevance

Table 3 shows an evaluation of the methodological quality of the studies selected according to the PEDro scale. Apart from the study of Fernández-de-las-Peñas et al. [28], which had a high methodological quality, the other studies [4,24,37] included in this review and meta-analysis were of low-quality.

The items that were least met in the methodology of these studies were: item 6 (there was blinding of all therapists who administered the therapy) and item 7 (there was blinding of all assessors who measured at least one key outcome).

The quality of the evidence of the included studies was assessed for seven outcomes using the GRADEpro development tool (Figure 5). Therefore, the validity of these results is limited due to age and gender differences. In addition, most studies demonstrated a risk of bias in more than one area in methodology. This fact coincides with the low methodological quality obtained with the PEDro scale.

## 4. Discussion

This study focuses on the efficacy of GPR in disease activity and mobility in AS. The systematic review and meta-analysis of the clinical trials included suggested that GPR was no more effective than other treatments in adults diagnosed with AS.

Among the studies examined, there was homogeneity of the samples in terms of sex, which is consistent with the literature [2]. They also present heterogeneity in the age of the participants (22 and 13–54 years) with very large standard deviations. Changes in AS are related to the evolution of the disease [38], and these, in turn, are related to age.

The criteria for the inclusion of participants in the trials analysed were homogeneous in terms of the rating scale used. All the studies used the criteria of the modified New York scale [25,30,38]. This scale is the most appropriate for the diagnosis of these patients. However, as indicated by Martins et al. [38], the evolution of the disease must be taken into account.

The GPR intervention in two of the trials was exactly the same [24,37] (GPR+McKenzie). Silva’s study was the only one to conduct a single GPR intervention. The remainder of the studies describe a treatment based on GPR but complemented by the McKenzie method in the extension-flexion motion of the lumbar spine. Coksevim´s and Durmus´ trials describe exactly the same treatment. However, in Coksevim´s trial, the intervention groups also received an anti-TNF drug treatment. Therefore, when we look at the results of the studies separately, they do not vary. In all interventions, the results were favourable. This suggests that the procedures we are describing are positive in the treatment of AS.

Conventional treatment was received by the control group in all studies reviewed. The results were mainly positive. This may be due to the fact that this treatment is based on movement, flexibility, and respiration; which are highly recommended in AS [12,15,38]. Therefore, conventional treatments and GPR are both adequate in the treatment of AS and the basic principles of both are the same.

The GPR is a physical therapy modality used by physiotherapists to improve axial mobility, functionality, chest expansion, and pain over conventional treatments [4,24,28,37]. These aspects are very important in AS patients. However, after the meta-analysis, we cannot affirm that it is a better therapeutic option than any other exercise, except for the finger-floor distance variable, in which case the GPR was better than conventional treatment. This variable is very relevant because it measures spinal mobility, a very important function in AS [1]. As for improvements in respiratory function, there were improvements in chest expansion, especially in combination with specific respiratory exercises [4,30].

One of the most relevant aspects of this study is that it provides a more specific view of the implementation of GPR focused on the AS. Thus, standard treatment of GPR has been described based on a minimum total treatment time of between 3 and 4 months (at one hour per week). This is divided into sessions in which, at least, a treatment posture is made (preferably stretching the posterior chain) combined with breathing exercises. The difficulty of the postures must be evolutionary, starting from the supine position to sitting or standing. The GPR sessions can be combined with conventional exercises or other methods, such as McKenzie, Pilates, or aerobic exercise [4,30]. The treatment should be supervised by a physiotherapist specialised in the method of intervention.

The small number of RCTs addressing this issue may be a limitation, which should be considered in further research.

The quality of the studies, according to the evaluation made with the PEDro scale, was low. We emphasize that the therapists who perform this type of GPR treatment cannot be blinded to either the intervention or the results of the patient’s evaluations. This is due to the peculiarities of the treatment and the basis of the method.

These results indicate that disease activity, as measured by BASDAI, improves with GPR [4,24,28,37]. This index has proven to be the best instrument to measure it. In addition, BASDAI is reliable in several population groups [33,34] for the measurement of the parameters of fatigue, spinal pain, joint pain/inflammation, pain in the prosthesis, and morning stiffness in AS. The BASFI index is the best tool to measure the functional capacity, health status, and evolution of patients diagnosed with AS [33,35]. Although Silva did not use it in his study, we propose that both instruments should be used in future studies.

## 5. Conclusions

GPR is beneficial for AS, but no more so than other conventional treatments, except for spinal mobility, as research observed that GPR improves the finger-floor distance. We suggest ongoing studies with good methodological quality in order to better clarify the usefulness of GPR for AS.

## Figures and Tables

**Figure 1 jcm-09-02696-f001:**
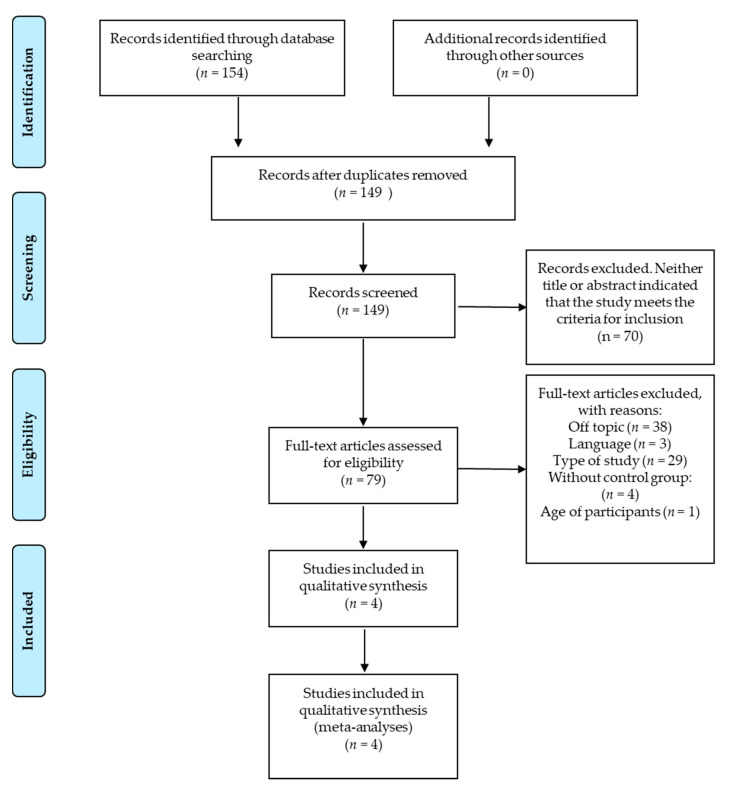
Flowchart of the articles selection process.

**Figure 2 jcm-09-02696-f002:**
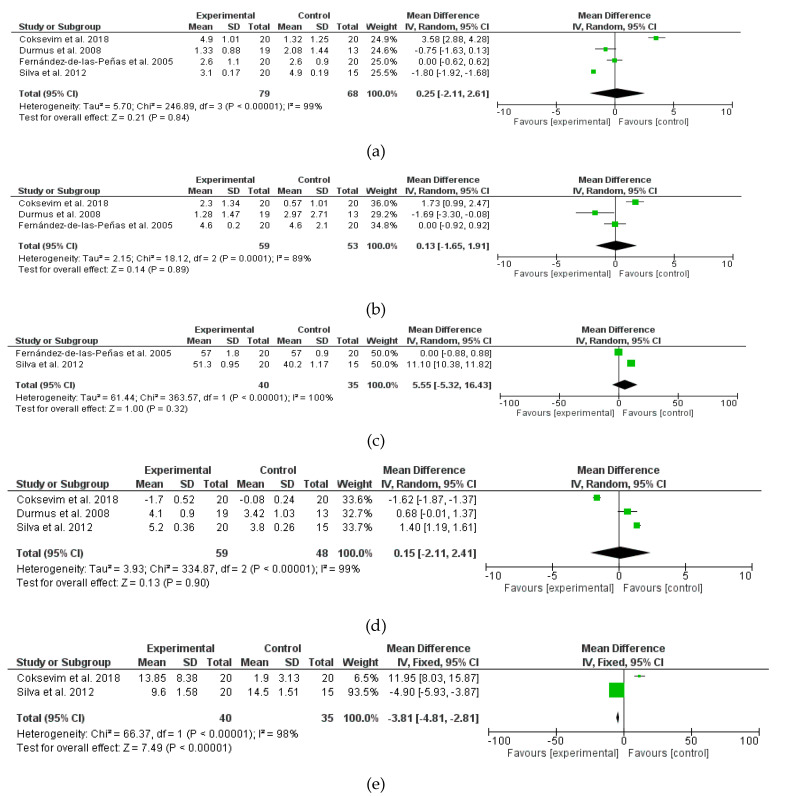
Multiple forest plot panels of the comparison of the difference between the groups in: (**a**) BASDAI; (**b**) BASFI; (**c**) cervical rotation; (**d**) chest expansion; (**e**) finger-floor distance; (**f**) modified Shöber test; and (**g**) VAS.

**Figure 3 jcm-09-02696-f003:**
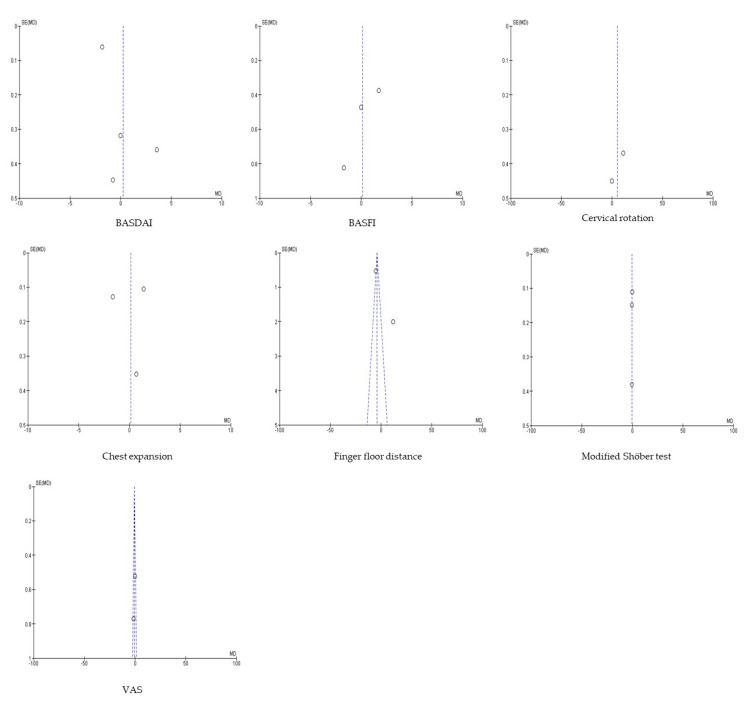
Multiple funnel plot panels.

**Figure 4 jcm-09-02696-f004:**
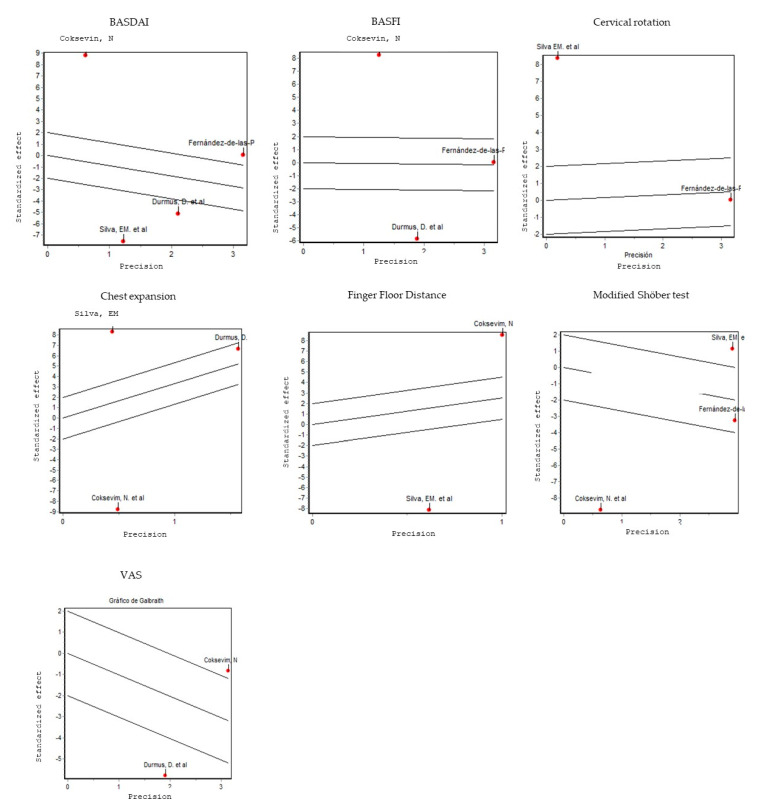
Galbraith graphs in: BASDAI; BASFI; cervical rotation; chest expansion; finger-floor distance; modified Shöber test; and VAS.

**Figure 5 jcm-09-02696-f005:**
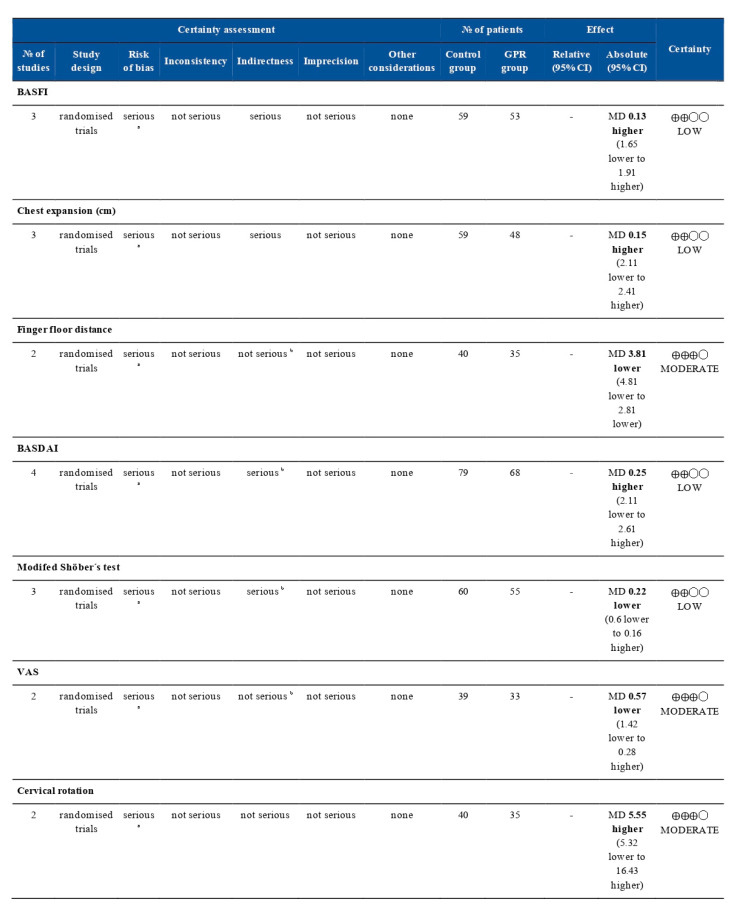
Quality evidence diagram obtained with the GRADEpro (Grading of Recommendations Assessment, Development, and Evaluation) development tool. CI: confidence interval; MD: mean difference. Explanations: ^a^ Most studies demonstrated a high risk of bias in more than one area in methodology; ^b^ Age and gender differences.

**Table 1 jcm-09-02696-t001:** Search Strategy.

Databases and Total Found Articles	Search
PubMed: 43 *PEDro: 14 *SciELO: 18 *WoS: 74	1. (“Pain”(Mesh) OR “Acute Pain”(Mesh) OR “Pelvic Girdle Pain”(Mesh) OR “Musculoskeletal Pain”(Mesh) OR “Chronic Pain”(Mesh) OR “Visceral Pain”(Mesh) OR “Nociceptive Pain”(Mesh) OR “Pain Perception”(Mesh) OR “Pain, Referred”(Mesh) OR “Shoulder Pain”(Mesh) OR “Neck Pain”(Mesh) OR “Pelvic Pain”(Mesh)) AND “Global Postural Reeducation” 2. (“Range of Motion, Articular”(Mesh)) AND “Global Postural Reeducation” 3. (“Quality of Life/psychology”(Major)) AND “Global Postural Reeducation” 4. (“Ventilation”(Mesh) OR “Respiration”(Mesh) OR “Ventilation-Perfusion Ratio”(Mesh) OR “Pulmonary Ventilation”(Mesh) OR “Maximal Voluntary Ventilation”(Mesh) OR “Respiration, Artificial”(Mesh)) AND “Global Postural Reeducation” 5. (“Spondylitis”(Mesh) OR “Spondylitis, Ankylosing”(Mesh) OR “Spondylarthropathies”(Mesh) spondyloarthritis ankylopoietica OR ankylosing spondylarthritis OR ankylosing spondylarthritides OR spondylarthritides, ankylosing OR spondylarthritis, ankylosing OR ankylosing spondylitis OR spondylarthritis ankylopoietica OR bechterew disease OR bechterew’s disease OR bechterew s disease OR marie-struempell disease OR marie struempell disease OR rheumatoid spondylitis OR spondylitis, rheumatoid OR spondylitis ankylopoietica OR ankylosing spondyloarthritis OR ankylosing spondylarthritides OR spondylarthritides, ankylosing OR spondyloarthritis, ankylosing) AND “Global Postural Reeducation” 6. “Global Postural Reeducation”

* PEDro, Physiotherapy Database; SciELO, Scientific Electronic Library Online; WoS, Web of Science.

**Table 2 jcm-09-02696-t002:** Clinical outcome data of the included studies.

Author (Year)	Intervention(s)	Control	Primary Outcome Measure	Secondary Outcome Measure	Follow-Up	Results
Coksevim, N et al., (2018) [24]	Anti-TNF therapy plus GPR program and conventional exercise therapy	Conventional exercises	BASDAI ^1^, BASFI ^2^, chest expansion, finger-floor distance, Modified Shöber test and VAS ^3^	6MWD ^4^, MAF ^5^, PSQI ^6^ and BDI^7^	3 months	The improvements in all parameters were better in the anti-TNF groups than in the control group regarding the change scores between BT ^15^-AT ^16^. The anti-TNF plus GPR exercise therapy resulted in greater improvements than the anti-TNF plus conventional exercise therapy in pain, walking performance, mobility parameters.
Durmus, D et al., (2008) [37]	Conventional exercise regimen and GPR	Conventional exercises	BASDAI ^1^, BASFI ^2^, Chest expansion and VAS ^3^	6MW ^4^, FVC ^8^, FEV1 ^9^, PEF ^10^, VC ^11^ and MVV ^12^	3 months	The intergroup comparison (pre-post scores) in both exercise groups showed that the GPR group obtained more improvement than the conventional exercise group in FVC, FEV1, and PEF parameters.
Fernández-de-las-Peñas, C et al., (2005) [28]	GRP	Conventional protocol of physical therapy in AS	BASDAI ^1^, BASFI ^2^, cervical rotation and modified Schöber test	Tragus to wall distance, lumbar side flexion, and intermalleolar distance	4 months	The intergroup comparison between the improvement (pre-post scores) in both groups showed that the GPR group obtained a greater improvement than the control group in all the clinical measures of cervical rotation and modified Schöber test lumbar side flexion, and intermalleolar distance, as well as in the BASFI index.
Silva, EM. et al., (2012) [4]	GPR	Conventional exercise	BASDAI ^1^, chest expansion and VAS ^3^	Morning stiffness, spine mobility, HAQ- S ^13^ and SF-36 ^14^.	4 months	In the inter-group comparison, there was significantly more improvement in the GPR group in all measures, except for the finger-floor distance (*p* = 0.12).

^1^ BASDAI: Bath Ankylosing Spondylitis Disease Activity Index; ^2^ BASFI: Bath Ankylosing Spondylitis Functional Index; ^3^ VAS: Visual Analogue Scale; ^4^ 6MWD: 6 min walk distance; ^5^ MAF: Multidimensional Assessment of Fatigue; ^6^ PSQI: Pittsburgh Sleep Quality Index; ^7^ BDI: Beck Depression Inventory; ^8^ FVC: Forced Vital Capacity; ^9^ FEV1: Forced Expiratory Volume in 1 s; ^10^ PEF: Peak Expiratory Flow; ^11^ VC: Vital Capacity; ^12^ MVV: Maximal Voluntary Ventilation; ^13^ HAQ-S: Health Assessment Questionnaire–Spondyloarthropathies–; ^14^ SF-36: Medical Outcome Study Short Form 36 Healthy Survey; ^15^ BT: Before treatment; ^16^ AT: After treatment.

**Table 3 jcm-09-02696-t003:** Evaluation of the quality of studies according to the PEDro scale.

Evaluation Criteria (Items) ^1^	1	2	3	4	5	6	7	8	9	10	11	Total Score
**Author-Year**												
**Coksevim, N. et al.-2018**	1	0	0	1	0	0	0	1	1	1	0	4
**Silva, E. et al.-2012**	1	0	0	1	0	0	0	1	1	1	0	4
**Durmus, D. et al.-2008**	1	0	0	0	0	0	0	1	1	1	1	4
**Fernández-de-las-Peñas, C. et al.-2005**	1	1	0	1	0	0	1	1	0	1	1	6

Score 0: the criterion is not met. Score 1: the criterion is met. ^1^ 1. Eligibility criteria were specified. 2. Subjects were randomly allocated to groups. 3. Allocation was concealed. 4. The groups were similar at baseline regarding the most important prognostic indicators. 5. There was blinding of all subjects. 6. There was blinding of all therapists who administered the therapy. 7. There was blinding of all assessors who measured at least one key outcome. 8. Measures of at least one key outcome were obtained from more than 85% of the subjects initially allocated to groups. 9. All subjects for whom outcome measures were available received the treatment or control condition as allocated or, where this was not the case, data for at least on key outcome were analysed by the “intention to treat”. 10. The results of between-group statistical comparisons were reported for at least one key outcome. 11. The study provided both point measures and measures of variability for at least one key outcome.

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
