# Peer review of "Effectiveness of Global Postural Reeducation in Ankylosing Spondylitis: A Systematic Review and Meta-Analysis"

_jcm, 2020, doi:10.3390/jcm9092696_

Round 1
Reviewer 1 Report
I read with interest this manuscript, which aimed to evaluate the effect of Global Postural Reeducation in patients with Ankylosing Spondylitis through the use of a systematic literature review and meta-analysis. The methodology is correct and follows the PRISMA recommendations.
However, I have some comments to the authors:
1. There are some misconceptions in the introduction section:
- "bamboo spine" does not represent an erosive inflammatory osteopenia; in fact, it is produced by new bone formation on the axial enthesis.
- The frequent age of onset is not 45 to 65 years. The disease usually starts before the age of 45 and, in the case of ankylosing spondylitis, this age is earlier.
- In the introduction section, authors stated "Recently, a strong genetic association between the HLA-B27 allele and AS has been reported". Thas association was described for the first time in the 70's. Please change the sentence.
2. Please, include the references of BASDAI, BASFI and BASMI in the section "2.5. Data analysis and outcomes".
3. In the flow chart, it would be interesting to specify the reason of exclusion of the 70 records during the screening.
4. In "3.1. Data extraction" section, first paragraph, please, explain what does "G" mean.
5. In "3.2. Data analysis and meta analysis", authors state "Seven meta-analyses were carried out, which can be simplified into two: one related to mobility and the other to pain". In do not understand this. The outcomes includes 4 domains: disease activity, function, mobility and pain. Please explain this.
6. The english language should be reviewed by a native speaker.
Reviewer 2 Report
Thank you for the opportunity to review this manuscript. Although in general terms I consider that a good work has been done, there are one major and several minor aspects that should be answered before considering the publication of this study.
It is necessary to make a better connection between the first three paragraphs of the introduction and the rest of it.
Line 32: A reference is missing after "bamboo spine".
Line 43: Absence of reference.
Lines 44-50: Lack of references for the theoretical foundation.
Line 128: Was not a cut-off point established to exclude studies based on their methodological quality?
Line 129: What evaluation instrument was used to evaluate the level of evidence and recommendation? Please indicate it.
In the flowchart it is indicated that 38 articles were excluded after reading the full-text, however, when detailing the reasons they only indicate 37. Please specify what reason explains the exclusion of the missing study.
In turn, if you had 79 articles for reading the full text and, after this, 38 were excluded, what happened to the remaining 41 during the critical analysis so that only four were included in the systematic review? This aspect questions the rest of the work done in this systematic review until this is resolved.
Line 143-144: What do the Gs refer to? Do I have to assume they are groups because of what is explained in the next line? On what criteria are these three groups formed? Later they are not mentioned again since they are divided into experimental and control.
Table 2: If you add the acronym in addition to the number at the foot of the table, perhaps it would be easier to identify them. Only as a recommendation.
Table 3: It is repetitive to show the PEDro scale score in both Table 3 and 4. In turn, the content of Table 3 is specified in text and later analyzed in depth in the discussion. Value the exclusion of it from the article.
Lines 227-245: The methodological quality assessment for the inclusion of the studies in the systematic review should be previously incorporated; possibly, before beginning to analyze in depth the characteristics and results of the studies.
Round 2
Reviewer 1 Report
My questions have been well addressed.
Author Response
Thank you very much for your help with the revision of this manuscript.
Reviewer 2 Report
In the flowchart there are still 36 studies that it is not known at what point in the process or for what reason they have been discarded. Please clarify this.
Author Response
Point 1: In the flowchart there are still 36 studies that it is not known at what point in the process or for what reason they have been discarded. Please clarify this.
Response 1: Thank you very much at this point. The flowchart was be modified according to the suggestion.
Thank you very much for your help with the revision of this manuscript.
Round 3
Reviewer 2 Report
Thank you for solving and answering all the suggestions made, congratulations on the work done.